# Experimental and Theoretical Approaches of New Nematogenic Chair Architectures of Supramolecular H-Bonded Liquid Crystals

**DOI:** 10.3390/molecules25020365

**Published:** 2020-01-16

**Authors:** O. A. Alhaddad, H. A. Ahmed, M. Hagar

**Affiliations:** 1Chemistry Department, College of Sciences, Taibah University, Madina Monawara 30002, Saudi Arabia; alhaddad105@yahoo.com; 2Chemistry Department, College of Sciences, Taibah University, Yanbu 30799, Saudi Arabia; 3Department of Chemistry, Faculty of Science, Cairo University, Cairo 12613, Egypt; 4Chemistry Department, Faculty of Science, Alexandria University, Alexandria 21321, Egypt

**Keywords:** chair-shaped supramolecular liquid crystals, hydrogen bonding, azopyridines, nematic phase, DFT theoretical calculations, molecular geometry

## Abstract

New four isomeric chair architectures of 1:1 H-bonded supramolecular complexes were prepared through intermolecular interactions between 4-(2-(pyridin-4-yl)diazenyl-(2-(or 3-)chlorophenyl) 4-alkoxybenzoates and 4-n-alkoxybenzoic acids. The H-bond formation of all complexes was confirmed by differential scanning calorimetry (DSC) and Fourier-transform infrared spectroscopy (FTIR). Mesomorphic characterization was carried by DSC and polarized optical microscopy (POM). It was found that all prepared laterally chloro-substituted supramolecular complexes were nematogenic, and exhibited nematic phase and low melting temperature. The thermal stability of the nematic mesophase observed depends upon the location and spatial orientation of the lateral Cl^−^ atom in as well as the length of terminal chains. Theoretical calculations were carried out within the paradigm of the density functional theory (DFT) in order to establish the molecular conformation for the formed complexes and estimate their thermal parameters. The results of the computational calculations revealed that the H-bonded complexes were in a chair form molecular geometry. Additionally, out of the acquired data, it was possible to designate the influence of the position and orientation of the lateral group as well as the alkoxy chain length on the stability of the nematic phase.

## 1. Introduction

Recently, supramolecular liquid crystals (SMLCs) attracted a vivid attention of the scientific community [1,2,3,4,5]. These systems combine the supramolecular chemistry [6] and liquid crystals [7,8] with efficient properties for optical and technological potential applications [9]. H-bonding intermolecular interactions are a well-established strategy to design self-assembly LCs through several non-covalent bonds [10,11,12,13,14]. Among the hydrogen bond acceptors and donors, the pair of a carboxylic acid and a pyridine derivative is the best choice in several studies. Moreover, the use of multifunctional components in the formation of a non-covalent interaction can produce better characteristics of supramolecular LC network architectures [6,7]. The supramolecular systems could be photosensitive host–guest complexes and they are of significant interest where the light can be applied in a remote manner as an external stimulus and offers excellent control with the wavelength [15,16,17]. Azopyridine molecules are incorporated into liquid-crystal materials to make them photoresponsive [18,19] due to their ability for trans–cis-isomerization upon thermal and photo-irradiation. Modifying the core structure or adding lateral substituents to azopyridine-based derivatives can lead to marked changes in photophysical and photochemical properties. [18,19] An incorporation of lateral groups with different size and polarity widely improves many characteristics of liquid crystalline materials. It could be attributed to the disturbance in the molecular packing that decreases the melting temperature and thermal stability of liquid crystal mesophases [20,21,22,23,24,25,26,27]. Lately, azopyridines have been used in the formation of nano-fiber supramolecular self-assembling and hydrogen/halogen-bonding LCs with photo-induced transition phenomena [28,29,30,31,32]. Designing of photosensitive SMLCs through intermolecular interactions using the suitable H-bond donors and acceptors are concerns of our area of interest [33,34,35,36,37,38,39]. Anisotropic structures are produced from the overall molecular shape of architectures and the combination of rigid (aromatic) and flexible segments (alkyl chains). Changes in the structure of molecules forming liquid crystalline phases impacts the mesomorphism as well as the properties essential for technical uses. Recently, construction of materials according to computational prediction has a high attention of many researchers [21,40,41,42,43,44,45,46,47,48,49]. Mutual influence of the many optical parameters requires stimulated information about the energies of molecular orbitals as well as the molecular geometries of the LCs. Moreover, density functional theory (DFT) is a powerful tool for taking an insight into features of the molecules at ease [21,43,50,51,52,53,54,55,56].

The goal of the present work was focused on designing the new series of liquid crystalline forming H-bonded supramolecular architectures and examines their physicochemical properties. Also, [55] to study the stability of different spatial oriented lateral polar groups on the thermal and optical behavior of prepared intermolecular H-bonded complexes, which oriented with different angles on the central ring of the azopyridine-based moiety, Scheme 1. Moreover, DFT theoretical calculations will be discussed to predict the molecular conformation for the formed complexes as well as their thermal parameters. In addition, these calculations will be used to explain the effect of the position and orientation of the lateral group as well as the length of the alkoxy chain on the type and the stability of the observed mesophase. Finally, to investigate the impact of the estimated thermal parameters of H-bonded complexes and how these parameters could affect their thermal and optical properties.

## 2. Results and Discussion

### 2.1. FT-IR Spectroscopic Confirmation of SMHB Complexes Formation

The formation of the supramolecular complexes has been confirmed by Fourier-transform infrared spectroscopy (FTIR) spectral data. The measurements were performed for the individual components as well as their H-bonded supramolecular complexes. The FTIR spectrum of acids, azopyridine bases and their complexes (**A***12***/I***16* and **A***12***II***16* as representative examples) are given in Figure 1. It has been reported that, no significant effect of the length of the alkoxy chain on the wavenumber of the C=O group stretching vibration either for the individual acids or the H-bonded complexes [39,57,58]. The signal at 1678.2 cm^−1^ was assigned to the stretching vibration of the C=O group of the alkoxy acid, experimentally and theoretically, respectively. The H-bonding between the nitrogen of azopyridines and the carboxylic group of alkoxybenzoic acid of the supramolecular complexes **A***n***/I***m* and **A***n***/II***m* replaces the bis H-bonds of the dimeric form of the alkoxybenzoic acid. One of the most important evidences of the H-bonded supramolecular complexes formation is the stretching vibration of the C=O carboxylic group either experimentally or theoretically. The sharing of carboxylic group OH-group in H-bonding formation will decreases the strength O-H bond. Theoretically, (Table 1), the OH-bond length increased from 0.97588 Å for the free acid to 1.04046 Å and 1.03154 Å for H-bonded complex **A***12***/I***16* and **A***12***/II***16*, respectively. Moreover, the wave-number of their stretching vibration decreases from 3660.9 cm^−1^ of the free acid to 2508.8 cm^−1^ for isomer **A***12***/I***16* and 2572.5 cm^−1^ for the other isomer, **A***12***/II***16*. Similarly, the strength of the C=O bond of the COOH group decreases upon the H-bonding formation, where, the stretching vibration decreases to 1687.0 and 1666.6 cm^−1^ for H-bonded isomers **A***12***/I***16* and **A***12***/II***16* instead of 1691.0 cm^−1^ for the free acid. Obviously, from the theoretical results, the position of the Cl-atom has an intensive effect on the H-bond strength of the H-bonded complex. The presence of the electronegative Cl-atom near the pyridine ring responsible for the H-bond formation for **A***12***/I***16* complex (the Cl-atom in meta position with respect to the ester group) will disrupt the H-bond formation by decreasing the availability of the lone pair on the N-atom of the pyridine ring. Experimentally, the results of the FT-IR revealed that no significant effect of the H-bond formation on the C=O group of the free carboxylic acid, only 2 cm^−1^ decreasing, (ύ_C=O_ = 1681.7 cm^−1^). However, the supramolecular complex formation has a high stretching vibration effect on the C=O of the ester linkage of the azopyridine base, their wave number increases from 1727.8 to 1743.4 cm^−1^ for complex **A***12***/I***16* and 15.9 cm^−1^ for the other complex **A***12***/II***16*. Moreover, it has been reported [49,59,60,61,62,63,64] that a major piece of evidence for the formation H-bond supramolecular complex is the presence of three vibration bands of Fermi resonance of the H-bonded OH groups **A**-, **B**- and **C**-types. The vibrational peak assigned to **A**-type Fermi band of complex **A***12***/I***16* and **A***12***/II***16* presented under the C-H vibrational peaks at 2915 to 2855 cm^−1^. Moreover, the peak at 2329 (**A***12***/I***16*) and 2356 cm^−1^ (**A***12***/II***16*) could be attributed to the O–H in-plane bending vibration as well as its fundamental stretch (**B**-type). However, 1899.3 and 1906.8 cm^−1^ were assigned to **C**-type Fermi band due to the interaction between the overtone of the torsional effect and the fundamental stretching vibration of the OH.

### 2.2. Mesomorphic and Optical Behavior

All 1:1 molar ratio complex, **A***n***/I***m* and **A***n***/II***m*, were prepared from two homologs of the azopyridine base (**I***m* and **II***m*) and four homologs of the acid, **A***n*. The prepared complexes were characterized by DSC and POM. The textures observed by POM were verified by the DSC measurements and types of mesophases were identified for all prepared supramolecular complexes **A***n***/I***m* and **A***n***/II***m*. DSC thermograms of the 1:1 supramolecular complexes **A***12***/I***16* and **A***12***/II***16*, as examples, are depicted in Appendix A. DSC behavior was observed for the prepared mixtures by subjecting them to repeat heating/cooling cycles.

Transition temperatures and their associated enthalpies of transition values were measured by DSC for all prepared complexes and are summarized in Table 2. The effect of terminal alkoxy chain length of the acid component (*n*) represented graphically, as a function of *m* of the two isomeric groups of base moieties (**I***m* and **II***m*) in Figure 2 and Figure 3, respectively. The results of Table 2 and Figure 2 and Figure 3 showed that neither terminal alkoxy chains of acid nor the base, *n* and *m*, respectively, effected the type of the mesophase, (nematic (N)), observed for all prepared lateral Cl complexes. In most cases, the N phase stability (***T***_N-I_) was found to decrease with the increment of *n*. As shown from Figure 2, the complexes **A***n/***I***8* exhibit an enantiotropic nematic phase and the nematic enhancement is slightly increases with the increment of *n* (Figure 2a). While, the longer base terminal (*m* = 16, Figure 2b), the prepared complexes **A***n/***I***16* showed a stable nematic phase upon heating and cooling except for **A***12/***I***16* exhibits monotropic N phase behavior. Upon heating, **A***12/***I***16* converts to isotropic liquid at 80.5 °C without showing any LC phase, whereas, in the cooling scan it exhibits a nematic mesophase start from 71.5 °C.

Figure 3 shows the mesomorphic behavior of base moiety **II***m* (the lateral Cl group introduced at the meta-position with respect to the ester carbonyl core) with variable alkoxy chains. It could be seen from Figure 3a, the supramolecular complexes **A***n/***II***8* exhibit different nematic behaviors than the corresponding isomeric complexes **A***n/***I***8,* whereas, **A***n/***II***8* have relatively wide enantiotropic nematic ranges with higher value for the complex **A***6/***II*8*** (~36.4 °C) and the wide nematic range value for **A***12/***II***8* monotropically. Moreover, the nematic stability decreases with the alkoxy chain length (*n*) of the acid component. In addition, the supramolecular complexes melting temperature is slightly affected by the length of the alkoxy chain of the acid. Finally, it is obvious from Figure 3b (**A***n/***II***16*) that, an independent effect of the alkoxy chain length of the acid on a monotropic nematic phase covered all supramolecular complexes. From the present investigation, it would be expected that the increment of the molecular anisotropy due to the orientation of the lateral electron-withdrawing Cl atom in the supramolecular geometry impacted the stability of nematic phase that agrees with our previous work [34,65].

Furthermore, the addition of lateral Cl atom in supramolecular architectures weakens the side by side cohesion interactions thus enhances a nematic phase for all 1:1 complex. Furthermore, the molecular geometry and size of the lateral substituent impact the mesophase stability and the polarizability of the whole molecule [22,23,66]. It was found that the length of the alkoxy chain, the polarity as well as the position (or orientation) of the lateral group are important factors determining the type and the range of the mesophase. Images of the mesophase as representative examples from POM are shown in Figure 4. Schlieren texture of the nematic phase was observed for all prepared complex.

### 2.3. Effect of Polarity and Orientation of Lateral Substituent on the Supramolecular Hydrogen-Bonded Complexes Stability

In order to study the effect of the polarity and the position (spatial orientation) of the lateral group on the mesophase thermal stability (**T**_C_) of 1:1 the prepared supramolecular H-bonded complex, a comparative study was constructed between mesophase stabilities (clearing temperature, ***T*_C_**) of the present lateral Cl complexes (**A***n***/I***m* and **A***n***/II***m*) and their corresponding lateral CH_3_ supramolecular H-bonded complexes (**A***n***/III***m* and **A***n***/IV***m*) [8,67], as well as the laterally neat one (**A***n***/V***m*) [36]. The data are represented graphically in Figure 5a–d. It had been found that the location and the inductive effect of the lateral substituent incorporated in base complement impact the polarizability between H-donors and H-acceptor and thus affect the strength of the H-bond [59]. However, the polarity of both components was not affected by the length of the terminal alkoxy chain (Figure 5a–d). Moreover, the laterally neat supramolecular H-bonded complexes (**A***n***/V***m*) have the highest thermal stability with respect to the derivatives of electron-donating CH_3_ and electron-withdrawing Cl lateral substituents. In addition, the nematic mesophase in the present investigation (lateral Cl complexes, **A***n***/I***m* and **A***n***/II***m*) is observed instead of the smectic C of the lateral CH_3_ and neat supramolecular complexes. Thus, the nature of intermolecular interactions between molecules affects the stability as well as the type of the mesophase. The lateral electron withdrawing Cl-atom of the complexes **A***n***/I***m* and **A***n***/II***m* predominates the end to end interaction to enhance a less ordered nematic phase, while the strong backing side by side interactions in the case of lateral CH_3_ (**A***n***/III***m* and **A***n***/IV***m*) and laterally neat (**A***n***/V***m*) complexes to give more ordered mesophase, smectic C, Scheme 2.

### 2.4. DFT Calculations

#### 2.4.1. Relationship between Experimental and Theoretical Parameters

The theoretical DFT calculations were performed in the gas phase by DFT/B3LYP method at 6-31G (d,p) basis set. All optimum compounds are stable, and this is approved in the term of the absence of the imaginary frequency. The results of the theoretical DFT calculations for lateral complexes of ortho chloro-derivatives with respect to the ester group (**A***n*/**I***m*) as well as the other isomeric supramolecular complex (meta-chloro with respect to the ester group) **A***12*/**II***16* and **A***16*/**II***16* showed a chair geometry for all investigated compounds. The three phenyl rings (two of the azopyridine base and one of the 4-alkoxybenzoic acid) of the H-bonded complexes are completely planar for both supramolecular H-bonded complexes. Recently, our group reported that [39], the chair forms conformation do not permit a strong lateral interaction leaving the end to end aggregation of the chains to be the predominant interaction. The pronounced terminal interaction could be a good explanation for the enhancement of the nematic mesophases observed for all alkoxy chain lengths of the H-bonded complexes over the parallel interaction that enhances the smectic phase formation, Figure 6. The estimated DFT calculations for thermal parameters, dipole moment and the polarizability of the prepared supramolecular hydrogen bonding liquid crystal complexes **A***12*/**I***16* and **A***n*/**II***m* are summarized in Table 3.

As shown from Table 3 and Figure 6, the length of the alkoxy chain of the homologs series enhancement the calculated thermal energy. As the chain length increases more packing of the molecules is permitted and consequently, the stability of the molecules increases [21,39,50,52,57,61,68]. Obviously, there is no significant effect of the alkoxy chain length on the dipole moment. However, the position and the spatial orientation of the Cl-atom has a high impact on the magnitude of the dipole moment, 6.8408 and 8.8598 Debye for ortho (**A**1*2*/**I***16*) and meta (**A**1*2*/**II***16*) chloro with respect to the carboxylate linkage, respectively. On the other hand, Figure 7 illustrates the relationship between the alkoxy chain length of acid moiety (*n*) and the polarizability. As the chain length increases the polarizability increases, and so, the candidate of the highest chain length showed the maximum polarizability and could be predicted to have the best characteristics in NLO applications. Moreover, the position and the orientation of the chloro-atom affects the predicted stability as well as the polarizability, the ortho chloro-derivative with respect to the ester group (**A***n*/**I***m*) showed higher polarizability and lower stability rather than that of the other isomer (**A***n*/**II***m*), the difference was 38.4 Bohr^3^ and 424.36 Kcal/mole, respectively, for *n* = 12, *m* = 16. The higher stability of the ortho chloro-derivatives could be illustrated in the term of its high degree of interaction of the molecules which permits more packing of the compounds rather than that of the meta derivatives.

Figure 8 shows the relationship between the length of the acid alkoxy groups and the mesophase nematic stability of 1:1 mixture **A***n*/**I***m* against the calculated thermal energy (**E_tot_**) and the polarizability (**α**). As shown from the figure, the length terminal alkoxy chain has a high effect on the mesophases stability of the nematic phase. The calculated thermal energy decreases with the length of the chain and mesophase stability decreases, the similar behavior was noticed with polarizability. The mesophase stability gradually decreases with the chain length up to *n* = 10 then sharp decrements were observed either with the estimated energy or polarizability. This result could be attributed to the high degree of the terminal aggregation at shorter chain lengths rather than that of the longer one which permits more parallel. The chair conformer structure of the H-bonded supramolecular compounds under investigation could permit the maximum end to end interaction for shorter chain lengths while for the longer one this interaction could be decreased with enhancement of side-side aggregation of alkoxy chains and the ester carbonyl moieties, that decreases the mesophase stability of the formed mesophase.

#### 2.4.2. Entropy Change of SMHB Complexes

Terminal alkoxy chains have a pronounced role as they are flexible and can easily make multi-conformational changes. An enhancement of the entropy change is observed in all supramolecular H-bonded complexes due to the increment in the conformational and orientation changes of the whole complex. A comparison of the normalized entropy changes for SMHB complexes **A***n*/**I***16*, **A***n*/**II***16*, **A***n*/**III***16*, and **A***n*/**IV***16* was depicted in Figure 9. Entropy of transitions (***∆S***/R) was constructed graphically as a function of the terminal alkoxy-chain length of acid component (*n*) for different lateral substituted supramolecular complexes. Figure 9 shows that, independent of the terminal flexible chains, an irregular entropy change was observed. That irregular change may be explained to the intermolecular interactions due to the location and rotation as well as the polarity of lateral substituent effect on the ordering of the whole complex [69,70]. The high dipole moment of **A***n*/**II***16* than **A***n*/**I***16* is accompanied by more conformational entropy changes due to good packing of lateral meta Cl supramolecular complexes molecules than the ortho Cl isomers. In contrast to the lateral electron-donating CH_3_ group, lower entropy transitions observed for meta CH_3_ SMHB complexes than the ortho CH_3_ isomeric complexes. These results could be explained in terms of the high degree of alignment of the molecules in the case of electron-donating lateral substituent (CH_3_) in the smectic mesophase that highly decreases the entropy with respect to the less ordered nematic mesophase in case of lateral electron-withdrawing group (Cl). The large value of entropy in many cases may be explained by the intermolecular interactions due to the location and rotation as well as the polarity of the lateral Cl-atom which enhancement the ordering of whole supramolecular complex. Moreover, non-correlation between the entropies and the terminal alkoxy-chain length may be due to the irregular change of lateral adhesion upon the increase of the total molecular length.

#### 2.4.3. Frontier Molecular Orbitals and Molecular Electrostatic Potential

Figure 10 summarizes the predicted ground state isodensity surface plots for the FMOs HOMO (highest occupied molecular orbital) and LUMO (lowest unoccupied molecular orbital)) as well as their energies difference (**ΔE**) of the compounds under investigation **A***n*/**II***m* and **A***12*/**I***16* as examples. As shown from Table 4, the FMO energy gap and the global softness (**S**) were not significantly affected by the length of the terminal alkoxy chain of compounds **A***n*/**II***m*. However, the position and the orientation of the lateral Cl atom have a high impact on the energy difference between the FMOs. The attachment of the Cl atom at the ortho position with respect to the ester linkage increases the energy difference between FMOs (HOMO and LUMO) than that at the meta position. This result could be helpful in the building of the molecules in a certain isomerism (positional and/or orientational) that would improve their characteristics to offer proper applications.

The charge distribution map for the complexes **A***12*/**I***16,*
**A***16*/**I***16* and **A***n*/**II***m* was calculated under the same basis sets according to molecular electrostatic potential (MEP) (Figure 11). The red region (negatively charged atomic sites) was distributed on the aromatic moiety and the maximum was carbonyl oxygen of the H-bonded carboxylic group, while alkoxy chains showed the least negatively charged atomic sites (blue regions). As shown in Figure 11, there is no significant effect of either the location, the orientation of the Cl atom or the alkoxy-chain length on the charge distribution. This could explain the reason of alteration of the type of the mesophase of the compounds under investigation in the term of the competitive interaction between end-to-end and side-side interaction by increasing of the chain length rather than the change of the charge distribution.

## 3. Experimental

### 3.1. Preparation of 1:1 Supramolecular Complexes

The 4-n-Alkoxy benzoic acids were obtained from Merck (Nuremberg, Germany). All the solvents used were of pure grade and purchased from Aldrich (St. Louis, WI, USA).

The 4-n-Alkoxy benzoic acids (**A***n*), and lateral chloro–pyridine-based derivatives (**I***m* and **II***m*) were checked to exhibit identical transition temperatures as previously reported [8,71]. The 1:1 molar ratio of any two complementary components SMHBLCs (Supramolecular H-bonded complexes) complexes (**A***n***/I***m* and **A***n***/II***m*) were prepared by melting the appropriate amounts of each component, stirring to give an intimate blend and then, cooling with stirring to room temperature (Scheme 3). For example, to prepare the supramolecular complex **A***10*/**I***8*: 0.0278 g of 4-decyloxybenzoic acid **A***10* and 0.0466 g of 4-(2-(pyridin-4-yl)diazenyl-(2-chlorophenyl) 4-octyloxy benzoate **I***8* were melted together to form the complex.

### 3.2. Characterizations

Supramolecular complexes formations were confirmed by TA Instruments Co. Q20 Differential Scanning Calorimeter (DSC; TA Instruments Co. Q20, DSC, New Castle, DE, USA), polarized-optical microscopy (POM, Wild, Humborg, Germany) and FT-IR (PerkinElmer, Inc., Shelton, CT, USA) spectroscopic analysis.

Calorimetric measurements were carried out using a PL-DSC of Polymer Laboratories, London, England. The instrument was calibrated for temperature, heat and heat flow according to the method recommended by Cammenga, et. al. [72] Measurements were carried out for small samples (2–3 mg) placed in sealed aluminum pans. All measurements were conducted at a heating rate of 10°C/min in an inert atmosphere of nitrogen gas (10 mL/min). For DSC, the sample was heated from room temperature to 280 °C at a heating rate of 10 °C/min under a nitrogen atmosphere, and then cooled in the cell to 0 °C. All weighed samples were made using an ultra-microbalance, Mettler Toledo, London, England, with accuracy ± 0.0001 mg.

Transition temperatures for the complexes (**A***n***/I***m* and **A***n***/II***m*) were investigated by DSC in heating and cooling cycles. The types of the mesophase were identified using standard POM (Wild, Germany), attached with Mettler FP82HT hot stage. Measurements were made twice, and the results were found to have accuracy in transition temperature and enthalpy within ± 0.2 °C.

### 3.3. Computational Methods and Calculations

The theoretical calculations for the investigated compounds were carried out by Gaussian 09 software [73]. DFT/B3LYP methods using a 6-31G (d,p) basis set was selected for the calculations. The geometries were optimized by minimizing the energies with respect to all geometrical parameters without imposing any molecular symmetry constraints. The structures of the optimized geometries had been drawn with Gauss view [74]. Moreover, the calculated frequencies were carried out using the same level of theory. The frequency calculations showed that all structures were stationary points in the geometry optimization method with none imaginary frequency.

## 4. Conclusions

Four new isomeric series of 1:1 SMHB complexes in chair-shaped liquid crystalline were constructed based on laterally Cl azopyridine derivatives and 4-alkoxybenzoic acids. All investigated complexes were confirmed by DSC, POM and FT-IR Fermi bands. It was found that all present 1:1 mixture is purely nematogenic with low melting temperatures. The experimental and DFT theoretical calculations results revealed that the H-bonded complexes were in a chair form molecular geometry. Moreover, the results of the DFT show that the position and orientation of the lateral group, as well as the alkoxy chain length, affects the type and the stability of the nematic mesophase. The position and the spatial orientation of the Cl-atoms have a high impact on the magnitude of the dipole moment as well as the polarizability. FMO energy gap and the global softness (**S**) were not significantly affected by the length of the terminal alkoxy chain of compounds. However, the position of the lateral Cl atom has a high impact on the energy difference between the FMOs. The higher stability of the ortho chloro-derivatives was illustrated in the term of its high degree of interaction of the molecules which permits more packing of the compounds rather than that of the meta derivatives. It could be concluded that the designing of new nematogenic supramolecular H-bonded conformers with certain molecular geometry offers suitable criteria that could be promising for proper optical applications. In addition, alteration of thermal and optical parameters by H-bonded complex formations and showing how could play an important role in improving the optical properties.

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
