# Peer review of "Experimental and Theoretical Approaches of New Nematogenic Chair Architectures of Supramolecular H-Bonded Liquid Crystals"

_molecules, 2020, doi:10.3390/molecules25020365_

Round 1

Reviewer 1 Report

In their manuscript Alhaddad et al. report the investigations of 1:1 molar ratio binary nematogenic mixtures. The compounds are characterized by a number of experimental techniques and by density functional theory calculations.

The reported results are of some interest for researches working in liquid crystal chemistry. The results could prove useful for the design of liquid-crystalline materials for technical applications. So the presented manuscript possesses some scientific merit. Unfortunately it is not clearly written. Some claims of the authors seem somewhat strange and unusual. Below I give some examples, but I must say that the manuscript in general needs a critical re-reading.

1.The authors write: “0.0278 mg of 4-decyloxybenzoic acid A10 and 0.0466mg of 4-(2-(pyridin-4-yl)diazenyl-(2-chlorophenyl) 4-octyloxy benzoate I8 were melted together” (lines 87-88). Did the authors really use such quantities of materials (on the order of 10-2 mg)?

 2. In line 134 and in Table 1 it is stated that the frequency of OH bond stretching vibration in A12 is 3660.9 cm-1. This seems to be a mistake.

3.What is the “mesophase stability temperature” (Figure 5)? I can guess this is the temperature of transition on heating from some mesophase into the isotropic phase. Am I correct?

4. In Conclusion the authors write: “It could be concluded that, the designing of new nematogenic supramolecular H- bonded conformers with certain molecular geometry offer a new phase transition phenomena that could be promising for proper optical applications”. What do the authors mean by “new phase transition phenomena”? It is not clear from the preceding text.

Although English is not my native language I have a strong impression that the English of the manuscript needs to be improved. Many of the sentences are unclear or seem erroneous, for example:

“supramolecular complexes were nematogenic exhibited pure nematic phase” (line 20)
“have an exponentially attraction in attention of scientific researches” (lines 38-39)
“This result could be help in building of the molecules in a certain isomerism (positional and/or orientational)” (lines 336-337)

I suggest that the authors consult a professional language editing service.

In my opinion a major revision of the manuscript is necessary.

Author Response

Dear Reviewer,

I would like first to thank the you for your valuable and accurate comments that helped us to revise the manuscript more thoroughly.  All your suggestions have been considered in the revised manuscript in a red color. 

1.The authors write: “0.0278 mg of 4-decyloxybenzoic acid A10 and 0.0466mg of 4-(2-(pyridin-4-yl)diazenyl-(2-chlorophenyl) 4-octyloxy benzoate Iwere melted together” (lines 87-88). Did the authors really use such quantities of materials (on the order of 10-2 mg)?

All written quantities are in gram not mg. It has been corrected.

In line 134 and in Table 1 it is stated that the frequency of OH bond stretching vibration in A12is 3660.9 cm-1. This seems to be a mistake.

This value is the calculated vibrational frequency of the OH group in the gas phase, not experimental value. For a single molecule in the gas phase there is no H-bonding and so it appears as a strong peak at 3660 cm-1.

3.What is the “mesophase stability temperature” (Figure 5)? I can guess this is the. Am I correct?

Yes, your guess is correct that is the highest temperature on heating at which transition from mesophase to the isotropic phase occurs.

In Conclusion the authors write: “It could be concluded that, the designing of new nematogenic supramolecular H- bonded conformers with certain molecular geometry offer a new phase transition phenomena that could be promising for proper optical applications”. What do the authors mean by “new phase transition phenomena”? It is not clear from the preceding text.

It has been corrected.

Although English is not my native language I have a strong impression that the English of the manuscript needs to be improved. Many of the sentences are unclear or seem erroneous, for example:

It has been checked.

“supramolecular complexes were nematogenic exhibited pure nematic phase” (line 20)
“have an exponentially attraction in attention of scientific researches” (lines 38-39)
“This result could be help in building of the molecules in a certain isomerism (positional and/or orientational)” (lines 336-337)

It has been checked.

Sincerely

Hagar

Reviewer 2 Report

The submitted article entitled "Experimental and theoretical approaches of new  nematogenic chair architectures of supramolecular H-bonded liquid crystals" by O.A. Alhaddad, H.A. Ahmed and M. Hagar presents the experimental results accompanied by DFT calculations for four new isomeric series of liquid-crystalline compounds based on laterally Cl azopyridine derivatives and 4-alkoxybenzoic acids.

It is a so-called "brick contribution", which does not bring much, yet it yields something new. Overall, the article is written on a mediocre level of scientific merit (severe mistakes were made) and the quality of language and the data presentation are far below acceptable. Despite the fact, that the role of the Reviewer is not to conduct language correction, I will enumerate some of them:

line 19: choro -> chloro lines 20-12: were nematogenic exhibited pure nematic phase and low melting temperature -> i.e. exhibited nematic phase and low melting
temperature lines 20-24: The density functional theory (DFT) theoretical calculations were discussed to predict the molecular conformation for the formed complexes as well as their thermal parameters -> Theoretical calculations were carried out within the paradigm of the density functional theory (DFT) in order to establish the molecular conformation for the formed complexes and estimate their thermal parameters lines 26-29:
Moreover, the results explained the effect the position and orientation of the lateral group as well as the alkoxy chain length on the type and the stability of the nematic mesophase. In addition, their impacts on the estimated thermal parameters of H-bonded complex and how these play an important role in influencing thermal and optical properties. -> Additionally, out of the acquired data it was possible to designate the influence of the position and orientation of the lateral group as well as the alkoxy chain length on the stability of the nematic phase. lines 38-39: Recently, supramolecular liquid crystals (SMLCs) have an exponentially attraction attention of scientific researches[1-5]. -> Recently, supramolecular liquid crystals (SMLCs) attracted a vivid attention of the scientific community. lines 58-59: Those changes in the characteristics of the LCs may be impact the mesomorphism as well as the properties essential for technical uses. -> Changes in the structure of molecules forming liquid crystalline phases impacts the  mesomorphism as well as the properties essential for technical uses. lines 62-64: Moreover, density functional theory (DFT) becomes effective popular method for its excellent performance and consistent with the experimental results. [18, 40, 47, 48] -> Moreover, density functional theory (DFT) is a powerful tool for taking an insight into features of the molecules at ease. lines 65-68: In order to understanding and controlling the mesomorphic properties of the soft material complexes, the goal of
present work focus on designing new H-bonded supramolecular architectures of new conformation and discuss the geometrical as well as the thermal parameters of the investigated complexes. -> The goal of the present work was focused on designing the new series of liquid crystalline forming H-bonded supramolecular architectures and examine their physicochemical properties. and the list continues...

Regarding the data presentation:

Figs. 1, 4, 6, 10 and 11 are of poor resolution. Fig. 4 could be enlarged. Fig. 6, 10 and 11 are hardly readable: enlarge and rearrange (the whole page could be utilized) Figs. 2, 3, 5, 7, 8, 9. What is the purpose of those lines? Are they guidance for the eye? Why splines and straight lines, is this some kind of fit? They should be justified in the relevant captions. Mesophase stability indicates range rather than a single point, thus either the plot should be reorganized or the axis caption changed. Tables 1, 3 and 4: round up to significant numbers

Severe mistakes:

The proper unit of temperature according to SI is Kelvin, yet authors use Celsius degree. Table 2 presents data from DSC. Enthalpy is given in kJ/mol and temperature in the Celsius degree.

According to the simple formula for entropy:

ΔS=ΔH/T,

where ΔH is enthalpy and ΔT temperature, one can fairly easy calculate ΔS based on the provided data. For example, see one for A6/I8 system (second row, fifth and sixth column). Putting the ΔH=1.66*103 (J/mol) and ΔT=350.45 K we get ΔS≈4.74 J/(mol*K). Onward, the dimensionless value ΔS/R gives approx. 0.56 which differs from one given in Table 2 (ΔS/R=2.58), which is in fact improper.

The authors acquired this improper value (and the rest presented in Table 2 regarding the ΔS/R) by dividing enthalpy by temperature in Celsius degree. Therefore it should be corrected.

Author Response

Dear Reviewer,

I would like first to thank the you for your valuable and accurate comments that helped us to revise the manuscript more thoroughly.  All your suggestions have been considered in the revised manuscript in a red color. 

line 19: choro -> chloro lines 20-12: were nematogenic exhibited pure nematic phase and low melting temperature -> i.e. exhibited nematic phase and low melting
temperature lines 20-24: The density functional theory (DFT) theoretical calculations were discussed to predict the molecular conformation for the formed complexes as well as their thermal parameters -> Theoretical calculations were carried out within the paradigm of the density functional theory (DFT) in order to establish the molecular conformation for the formed complexes and estimate their thermal parameters lines 26-29: 
Moreover, the results explained the effect the position and orientation of the lateral group as well as the alkoxy chain length on the type and the stability of the nematic mesophase. In addition, their impacts on the estimated thermal parameters of H-bonded complex and how these play an important role in influencing thermal and optical properties. -> Additionally, out of the acquired data it was possible to designate the influence of the position and orientation of the lateral group as well as the alkoxy chain length on the stability of the nematic phase. lines 38-39: Recently, supramolecular liquid crystals (SMLCs) have an exponentially attraction attention of scientific researches[1-5]. -> Recently, supramolecular liquid crystals (SMLCs) attracted a vivid attention of the scientific community. lines 58-59: Those changes in the characteristics of the LCs may be impact the mesomorphism as well as the properties essential for technical uses. -> Changes in the structure of molecules forming liquid crystalline phases impacts the  mesomorphism as well as the properties essential for technical uses. lines 62-64: Moreover, density functional theory (DFT) becomes effective popular method for its excellent performance and consistent with the experimental results. [18, 40, 47, 48] -> Moreover, density functional theory (DFT) is a powerful tool for taking an insight into features of the molecules at ease. lines 65-68: In order to understanding and controlling the mesomorphic properties of the soft material complexes, the goal of
present work focus on designing new H-bonded supramolecular architectures of new conformation and discuss the geometrical as well as the thermal parameters of the investigated complexes. -> The goal of the present work was focused on designing the new series of liquid crystalline forming H-bonded supramolecular architectures and examine their physicochemical properties. and the list continues...

All have been addressed according to the Referee's advice.

Regarding the data presentation:

Figs. 1, 4, 6, 10 and 11 are of poor resolution. Fig. 4 could be enlarged. Fig. 6, 10 and 11 are hardly readable: enlarge and rearrange (the whole page could be utilized)

It has been addressed

 Figs. 2, 3, 5, 7, 8, 9. What is the purpose of those lines? Are they guidance for the eye? Why splines and straight lines, is this some kind of fit? They should be justified in the relevant captions.

 In fact, these lines are important to show the relation trends and to be easy for the readers.

 Mesophase stability indicates range rather than a single point, thus either the plot should be reorganized or the axis caption changed.

It has been replaced by clearing temperature, Tc.

 Tables 1, 3 and 4: round up to significant numbers

They have been rounded up.

Severe mistakes:

The proper unit of temperature according to SI is Kelvin, yet authors use Celsius degree. Table 2 presents data from DSC. Enthalpy is given in kJ/mol and temperature in the Celsius degree.

According to the simple formula for entropy:

ΔS=ΔH/T,

where ΔH is enthalpy and ΔT temperature, one can fairly easy calculate ΔS based on the provided data. For example, see one for A6/I8 system (second row, fifth and sixth column). Putting the ΔH=1.66*10(J/mol) and ΔT=350.45 K we get ΔS≈4.74 J/(mol*K). Onward, the dimensionless value ΔS/R gives approx. 0.56 which differs from one given in Table 2 (ΔS/R=2.58), which is in fact improper.

The authors acquired this improper value (and the rest presented in Table 2 regarding the ΔS/R) by dividing enthalpy by temperature in Celsius degree. Therefore it should be corrected.

Yes, that is human mistake and it was corrected and tabulated in Table 2, Table S1 and Figure 9.

Sincerely

Hagar

Reviewer 3 Report

The manuscript of O.A. Alhaddad, H.A. Ahmed, M. Hagar describes H-bonded supramolecular architectures based on 1:1 complexes of laterally azopyridine derivatives and 4-alkoxybenzoic acids. Synthesized H-bonded complexes are well characterized; DSC, POM and FT-IR Fermi bands confirmed formation of complexes. It was found that all complexes are purely nematogenic with low melting temperatures. The experimental and DFT theoretical calculations results revealed that the H-bonded complexes were in a chair form molecular geometry. Moreover, the results of the DFT show that the position and orientation of the lateral group as well as the alkoxy chain length affects the type and the stability of the nematic mesophase.

I think that the manuscript is of a high quality; it adds new knowledge to the existing one on the H-bonded supramolecular architectures. The described method is an efficient approach for the synthesis of complexes of azopyridine derivatives with 4-alkoxybenzoic acids; the complexes could demonstrate interesting light induced switching ability that should be checked. After corrections of misprints and correcting English, I recommend acceptance of this manuscript.

Author Response

Dear Reviewer,

I would like first to thank the you for your valuable and accurate comments that helped us to revise the manuscript more thoroughly.  All your suggestions have been considered in the revised manuscript in a red color. 

I think that the manuscript is of a high quality; it adds new knowledge to the existing one on the H-bonded supramolecular architectures. The described method is an efficient approach for the synthesis of complexes of azopyridine derivatives with 4-alkoxybenzoic acids; the complexes could demonstrate interesting light induced switching ability that should be checked. After corrections of misprints and correcting English, I recommend acceptance of this manuscript.

In fact, our choice of these types of supramolecular complexes due to their relatively low melting points that can be used in many applicable fields. As suggested by the referee, the topic of light induced switching ability of these new supramolecular complexes will be discussed in our future perspectives in details.

Sincerely

Hagar

Round 2

Reviewer 1 Report

The authors have improved the presentation style of their manuscript. My questions and comments have been addressed properly. I can recommend the revised version of the manuscript for publication in Molecules.

The English of the manuscript has been improved considerably, however it seems several typos are still present in the text. Maybe one more re-reading of the manuscript would be helpful.

Author Response

Dear Reviewer,

I would like first to thank the you for our valuable and accurate comments that helped us to revise the manuscript more thoroughly.  The English have been considered in the revised manuscript in a red color. 

Sincerely

Hagar

Reviewer 2 Report

Authors have taken into account suggestions envisaged, yet very selectively.

Figures are still of poor quality (resolution) - Fig. 6, 10 and 11.

The intention of my comment:

"Figs. 2, 3, 5, 7, 8, 9. What is the purpose of those lines? Are they guidance for the eye? Why splines and straight lines, is this some kind of fit? They should be justified in the relevant captions."

was to force the authors not only to convince me but also the potential Reader. Unfortunately, neither I was convinced or will be the Reader (no explanation directly in the caption to relevant figure).

Authors write:

In fact, these lines are important to show the relation trends and to be easy for the readers.

What is the argument behind calling those lines "trends", where for example in Fig. 2a and 3b, one set of data is joined by straight lines alongside the second one, which is joined by splines (?). The authors blur the view.

That blurring also traverses into the unit presentation, like in lines 114 and 115. Would it not be more convenient to write the accuracy in μg?

Author Response

Dear Reviewer,

I would like first to thank you for our valuable and accurate comments that helped us to revise the manuscript more thoroughly.  All your suggestions have been considered in the revised manuscript in a red color. 

Figures are still of poor quality (resolution) - Fig. 6, 10 and 11.

The figures have been changed with good quality one.

The intention of my comment:

"Figs. 2, 3, 5, 7, 8, 9. What is the purpose of those lines? Are they guidance for the eye? Why splines and straight lines, is this some kind of fit? They should be justified in the relevant captions."

was to force the authors not only to convince me but also the potential Reader. Unfortunately, neither I was convinced or will be the Reader (no explanation directly in the caption to relevant figure).

Authors write:

In fact, these lines are important to show the relation trends and to be easy for the readers.

What is the argument behind calling those lines "trends", where for example in Fig. 2a and 3b, one set of data is joined by straight lines alongside the second one, which is joined by splines (?). The authors blur the view.

Regarding figure 2, it is known that the relationship between the Tc and the chain length is irregular trend so in the literature it is used to be drawn as splines. However, figure 3, we wants to show the effect of the lateral groups on the Tc with chain length and so we have drawn it as straight lines.

Concerning figure 7 and 8 we just connected without fitting the points to show the relations. However, we find splines for figure 8 will be more informative than straight lines.

Figure 9 is an irregular relationship between the change in the entropy with the chain length.

That blurring also traverses into the unit presentation, like in lines 114 and 115. Would it not be more convenient to write the accuracy in μg?

It was a personal error and we had corrected the unites to grams instead of mg.

Sincerely

Hagar